# Robust Secure Swap: Responsible Face Swap With Persons of Interest Redaction and Provenance Traceability

Yunshu Dai [1]   Jianwei Fei [2]   Fangjun Huang [1 3]   Chip Hong Chang [4]

## Abstract

As AI generative models evolve, face swap technology has become increasingly accessible, raising concerns over potential misuse. Celebrities may be manipulated without consent, and ordinary individuals may fall victim to identity fraud. To address these threats, we propose *Secure Swap*, a method that protects persons of interest (POI) from face-swapping abuse and embeds a unique, invisible watermark into nonPOI swapped images for traceability. By introducing an ID Passport layer, Secure Swap redacts POI faces and generates watermarked outputs for nonPOI. A detachable watermark encoder and decoder are trained with the model to ensure provenance tracing. Experimental results demonstrate that Secure Swap not only preserves face swap functionality but also effectively prevents unauthorized swaps of POI and detects different embedded model's watermarks with high accuracy. Our method achieves a 100% success rate in protecting POI and over 99% watermark extraction accuracy for nonPOI. Besides fidelity and effectiveness, the robustness of protected models against image-level and model-level attacks in both online and offline application scenarios is also experimentally demonstrated. Code can be found at https://github.com/SleepyCat888/Robust-Secure-Swap.

---

[1]School of Cyber Science and Technology, Shenzhen Campus of Sun Yat-sen University. [2]State Key Laboratory of Internet of Things for Smart City, Department of Computer and Information Science, University of Macau. [3]Guangdong Province Key Laboratory of Information Security Technology. [4]School of Electrical and Electronic Engineering, Nanyang Technological University.. Correspondence to: Fangjun Hung <huangfj@mail.sysu.edu.cn>.

*Proceedings of the $42^{nd}$ International Conference on Machine Learning*, Vancouver, Canada. PMLR 267, 2025. Copyright 2025 by the author(s).

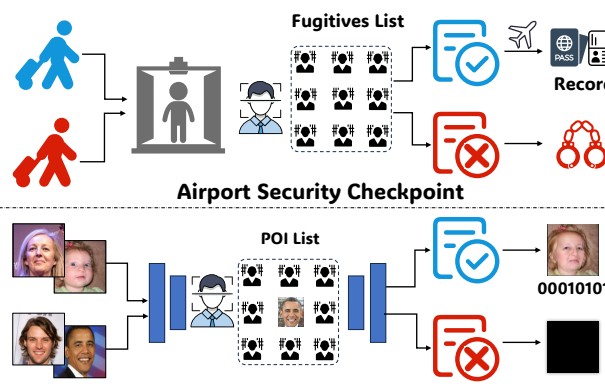

*Figure 1.* An illustration of our motivation behind *Secure Swap*. If the input image involves POI, the swap will be rejected; otherwise, the model will complete the swap and apply invisible watermarks in the generated images.

## 1. Introduction

In the self-media era, the general public has easier access to face swap technology. However, while this technology offers new possibilities, it has also raised significant concerns due to its potential for misuse. Users can provide the source and target faces in pairs into the face swap model The source image provides the identity (i.e., the face to be transferred), and the target image is the image where the person's identity is replaced by that of the source. Simply put, it uses the source person's face to replace the target person's face. Face swap operations targeting persons of interest (POI) can easily cause social unrest due to the sensitivity of their identities, while ordinary people (nonPOI) are more vulnerable to face swap fraud due to limited defense awareness and forensic capabilities (Van der Sloot & Wagensveld, 2022). To address these risks, different regions have proposed distinct governance frameworks for face swap technologies [1].

Currently, the growing application of generative technologies among the public has led to a more intricate and convoluted crisis. Open-source tools are becoming more accessible and easier to deploy for local inference, while API

---

[1]https://www.weforum.org/agenda/2023/11/generative-ai-governance-regulation

services are expanding at an unprecedented rate. Traditional defensive strategies, such as incorporating a front-end module within API or model, are no longer sufficient. As a result, the demand for robust and generalizable governance of face swap models have grown increasingly urgent.

Several studies have been conducted to counteract the abuse of face swap models, primarily through forensic detection (Dai et al., 2022; Fei et al., 2022a; Wang et al., 2023; 2022; Wu et al., 2023), which is applied to forged images after the abuse has already occurred. However, proactive defense measures during the model development stage, such as function purification (Dai et al., 2024a;b) and forensic collaboration, have been largely overlooked. Under a responsible governance framework, it is essential for model developers to take on the obligation of ensuring the legitimate use of their models, incorporating generative content security and forensic accountability.

To address these challenges, we propose **Secure Swap**, a method specifically designed for GAN-based face swap models. Secure Swap prohibits face swapping on POI and marks the synthesized products of nonPOI by embedding invisible watermarks for provenance tracking. Inspired by the concept of security checkpoints, our method integrates an identity checkpoint within the face swap model: if the input image involves a POI, the model redacts the face; otherwise, it generates a swapped image with an embedded invisible watermark. This approach ensures the ethical use of face swap technologies and reduces the risk of face fraud. The contributions of our work are summarized as follows:

- We propose a secure face swap model enhancement method called **Secure Swap** to enable any GAN-based face swap models to operate responsibly and with accountability. It redacts POI faces and ensures every swapped image of nonPOI carries an invisible watermark, simplifying real/fake detection and provenance tracking.

- Extensive evaluations on four face swap models and three human face datasets demonstrate that Secure Swap achieves a 100% success rate in protecting POI and greater than 99.9% watermark extraction accuracy, without compromising the visual quality of original outputs.

- Secure Swap is robust against various attacks, including image-level attacks, model-level attacks, and adversarial attacks. It incurs minimal training overhead and allows multiple instances of the same model to be traced with unique watermarks without additional training.

## 2. Related Work

Compared to passive defense (Fei et al., 2022c; Yang et al., 2023; Yu et al., 2022c; Zhao et al., 2023), proactive defense entails implementing preventive measures before the

occurrence of forgery, allowing for early response to potential forgery threats. At present, the most active defense measures are POI protection and watermark provenance.

**POI Protection.** Persons of Interest (POI) such as social celebrities and national leaders typically possess extensive social influence and receive high levels of attention. Their statements, actions, and images have a significant impact on the public, which is further amplified by the prevalence of social media. Therefore, forgers have a particular penchant for forging celebrities to disseminate false information, engage in scams, or perpetrate malicious activities. At present, a series of protection works for POI has emerged. Agarwal *et al.* (Agarwal et al., 2019) focused on the videos of talking in a formal setting like news interviews and public speeches. This work tracks face and head movements and then extracts the presence and strength of specific action units for POI DeepFake detection. Chu *et al.* (Chu et al., 2022) hypothesized that certain words or sentences spoken by POI tend to be accompanied by a series of regular face movements and lip motions. They adopt a dual stream structure for speaking pattern modeling, which combines the Action Unit Module that depicts face muscle movements and the Lip Motion Module that extracts sequential lip motions for detection. Likewise, Cozzolino *et al.* (Cozzolino et al., 2023) hypothesized that each person has specific characteristics that are unlikely to be reproduced by a synthetic generator. They extracted audio-visual features that characterize the identity and used them to create a POI DeepFake detector.

Existing methods to address unauthorized face swaps fall into two categories: proactive protection and post-hoc detection. Post-hoc detection suffers from inherent latency and cannot effectively prevent image misuse. Current proactive methods primarily rely on adversarial attacks by adding perturbations to protected images, such as POI photos, to prevent forgery (Huang et al., 2022; 2021; Ruiz et al., 2020). These approaches offer image-level protection, require costly preprocessing, and cannot scale to large volumes.

**Generative model watermarking.** Generative model watermarking has been explored to embed traceable information into outputs of GANs, ensuring copyright protection and supporting proactive forensics (Fei et al., 2022b; Yu et al., 2021; 2022b). For instance, Yu *et al.* (Yu et al., 2021) introduced a method to embed watermarks into training data, while others (Fei et al., 2022b; Wu et al., 2020) enhanced robustness against manipulations like compression and cropping. However, these methods are limited to post-forgery detection, whereas our approach proactively prevents identity misuse by instructing generative models to refuse forgery.

Unlike existing watermarking methods (training data based embedding (Yu et al., 2021), watermark decoder based supervised embedding (Fei et al., 2024; Fernandez et al.,

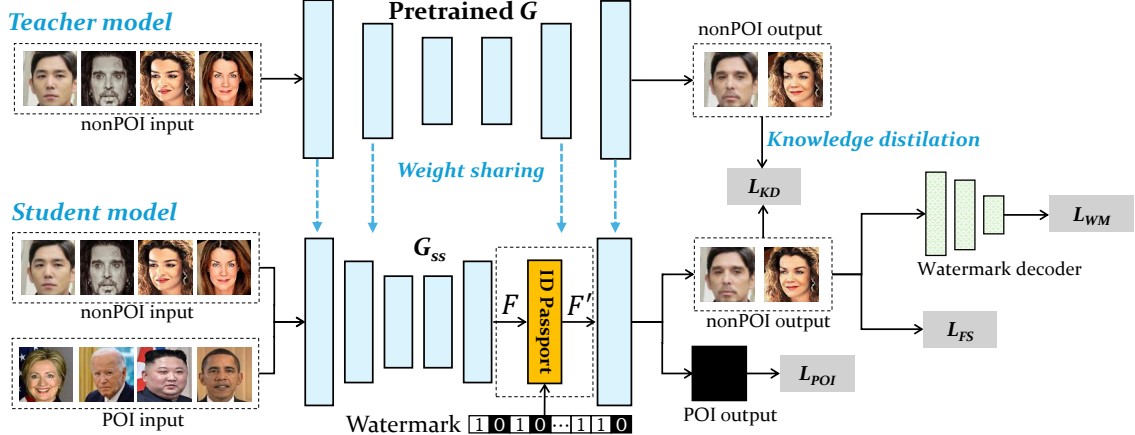

*Figure 2.* An overview of the training pipelines of the proposed *Secure Swap* method.

2023)), our method directly encodes the watermark into model parameters. Although the watermark is also extracted from images, it becomes part of the model parameters. Once initialized by a watermark, the model no longer requires external watermark input during inference. This supports efficient, scalable creation of uniquely watermarked model instances. Our design allows rapid deployment of distinct watermarked model instances at scale.

## 3. Methodology

We propose *Secure Swap*, a method enabling GAN-based face swap models to redact POI and embed invisible watermarks for provenance tracing, promoting the governance of responsible face-swapping technology. An illustration of this scenario is provided in the *Appendix*. Fig. 2 provides an overview of *Secure Swap*, which depicts the training pipelines involved in transforming a developer's face swap model $G$ into a released version $G_{SS}$ that incorporates security functions into the original face swap functionality. We first create a replicate model $G_{SS}$ with the same architecture and parameters as the developer's originally trained face swap model $G$. A small subnetwork with learnable parameters, called the *ID Passport* layer, is then inserted into $G_{SS}$. The ID passport layer is designed with the aim to black out the faces of POI outputs and embedding watermark features into the nonPOI outputs. Then, $G_{SS}$ is retrained with a dataset that contains the original dataset of nonPOI images and an additional set of POI images, all with labeled identities. Four loss functions are used simultaneously for the training of $G_{SS}$.

### 3.1. Face swap functionality preservation

In order to preserve the face swap functionality and visual quality of the originally trained model $G$ when instilling new security features into the secure swap model $G_{SS}$, we use $G$ to guide the training of $G_{SS}$. More specifically, we

use the Learned Perceptual Image Patch Similarity (LIPIPS) (Zhang et al., 2018) as a vehicle to distill the knowledge from $G$ to $G_{SS}$. We employ Knowledge Distillation (KD) to guide the teacher-student model framework, enabling the student model $G_{SS}$ to learn the non-POI outputs generated by the teacher model $G$. It ensures that the student model maintains visual quality and functionality comparable to the teacher model while incorporating the security features. The knowledge distillation loss is calculated based on the nonPOI outputs of the teacher model $G$ and the student model $G_{SS}$ to ensure that the swapped images generated by $G_{SS}$ is visually close to those of $G$. The knowledge distillation loss $\mathcal{L}_{KD}$ is given by:

$$\mathcal{L}_{\text{KD}} = \sum_l \frac{w_l}{H_l W_l} \sum_{h=1}^{H_l} \sum_{w=1}^{W_l} \big\| \phi_l(G(x_s, x_t))_{h,w} \\ - \phi_l(G_{SS}(x_s, x_t))_{h,w} \big\|_2^2, \quad (1)$$

where $G(x_s, x_t)$ and $G_{SS}(x_s, x_t)$ are the images generated by the teacher and student models, respectively with the same source image $x_s$ and target image $x_t$. $\phi_l(I)$ represents the activation of the $l$-th layer of a pre-trained network for the image $I$. $w_l$ is the learned weight of the $l$-th layer. $H_l$ and $W_l$ are the height and width, respectively of the $l$-th layer's activation map. $\|\cdot\|_2$ denotes the Euclidean $L_2$ norm.

In addition, we use the identity distance between the source and swapped images as a vehicle to regulate the training of $G_{SS}$. Let $x_s$, $x_t$ and $y_{s,t} = G(x_s, x_t)$ denote the source, target, and generated images, respectively. Following the previous research (Shiohara et al., 2023), we adopt ArcFace (Deng et al., 2019), a popular tool used in the field of face recognition and verification, to extract identity information for the identity distance calculation. The identity loss between a source image $x_s$ and swapped image $y_{s,t}$ is defined as follows:

$$\mathcal{L}_{\text{id}} = 1 - \cos(E_{\text{id}}(x_s), E_{\text{id}}(y_{s,t})), \quad (2)$$

where $E_{\text{id}}$ denotes ArcFace encoder and $\cos\langle u, v\rangle$ is the cosine similarity between vectors $u$ and $v$.

To ensure the student model $G_{SS}$ focuses only on changing the identity of the output to that of the source image without altering the background, expressions, garment, headdress and other contents of the target image, when the two different source and target images are sampled from the same identity, the swapped image should look perceptually similar to the target image. This case ($x_t$ and $x_s$ belong to the same identity) of face swapping is analogous to reconstruction. The reconstruction loss $\mathcal{L}_{\text{rec}}$ is calculated by the $L_1$ distance between the target image $x_t$ and the generated image $x_{s,t}$ as follows:

$$\mathcal{L}_{\text{rec}} = \begin{cases} \|x_t - y_{s,t}\|_1 & \text{if } ID(x_t) = ID(x_s), \\ 0 & \text{otherwise.} \end{cases} \quad (3)$$

Aside from enforcing the forward consistency of swapping images with the same identity, we also take into account the backward consistency of converting the swapped image back to the original target image. This is accomplished by applying cycle consistency loss $\mathcal{L}_{\text{cyc}}$ in (4) to 20% ($p = 0.2$) of nonPOI source and target images with the same identity when training the $G_{SS}$. The cycle consistency loss $\mathcal{L}_{\text{cyc}}$ is

$$\mathcal{L}_{\text{cyc}} = \|x_t - G(x_t, y_{s,t})\|_1. \quad (4)$$

The loss functions, (2), (3) and (4), are combined into a face swap loss for the nonPOI training of the $G_{SS}$. The face swap loss $\mathcal{L}_{FS}$ is obtained as

$$\mathcal{L}_{FS} = \mathcal{L}_{\text{id}} + \mathcal{L}_{\text{rec}} + \mathcal{L}_{\text{cyc}}. \quad (5)$$

### 3.2. POI redaction and provenance traceability

As mentioned earlier, the only difference between the architectures of $G$ and $G_{SS}$ is the ID Passport layer, which is added into the middle layer nearer to the output of $G_{SS}$. The input image propagates through multiple layers of $G_{SS}$ in the form of feature maps before reaching the ID passport layer. The ID passport layer converts the output feature map $F$ of its preceding layer to the feature map $F'$ that carries the identity and watermark features. $F'$ is output to the succeeding layer in $G_{SS}$, as shown in Fig. 2.

The general architecture of the ID passport layer is shown in Fig. 3. It consists of several parallel ID convolution layers, *IDConv*$_1$ to *IDConv*$_n$. Each kernel *Kernel*$_i$ of *IDConv*$_i$, $i = 1, 2, \cdots, n$, is derived from an upstream feature map $F$ through a kernel generator. Each *IDConv*$_i$ is also weighted by a coefficient $c_i$ generated by a coefficient generator from $F$, and a watermark feature vector $WM = [wm_1, wm_2, \cdots, wm_f]$ generated by a detachable watermark encoder from a binary watermark string $w$. During training, a different randomly generated $w$ is used for

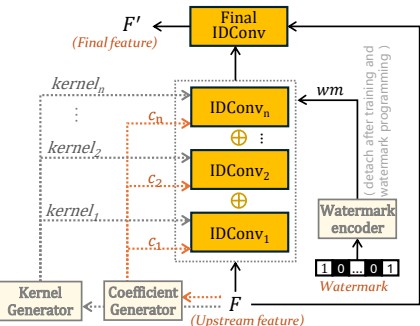

*Figure 3.* Architecture of ID passport layer with the input upstream feature $F$ and the output final feature $F'$.

each training batch. The watermark encoder is trained to generate different watermark feature vectors *WM* from $w$ and a watermark decoder (see Fig. 2) is simultaneously trained to extract $w$ embedded through $F'$ into the nonPOI generated face swap outputs. The combined output of all IDConv blocks is convoluted with $F$ to produce the final feature map $F'$ to the next layer of $G_{SS}$. When the training is completed, the model developer will input a unique $w$ into the trained watermark encoder to generate a fixed feature vector *WM* to program the *IDConv* in the passport layer of each trained $G_{SS}$ instance. The entire trained watermark encoder will be detached from the passport layer of the trained $G_{SS}$ instance before it is released to the customer or licensee. The trained watermark encoder is used to attach to the passport layer of any trained $G_{SS}$ instance to program a different $w$ into it. Details of the kernel generator, coefficient generator and watermark encoder involved in the construction of IDConv are provided in Fig. 4.

For POI outputs, to enforce $G_{SS}$ to output a black redacted image, the $L_2$ distance between the POI output image and a black image (denoted as $B$) is minimized by the following POI loss function:

$$\mathcal{L}_{\text{POI}} = \mathop{\mathbb{E}}_{x_s \sim X_s, x_t \sim \text{POI}} \|B - G_{SS}(x_s, x_t)\|_2^2, \quad (6)$$

where $X_s$ is a source image set.

For nonPOI outputs, a watermark decoder $D_w$ is trained with the Binary Cross-Entropy (BCE) loss to accurately extract the watermark from the nonPOI outputs. BCE loss is commonly used in binary classification tasks. It measures the difference between the predicted probabilities and the actual binary labels, and is calculated by:

$$\begin{aligned} \mathcal{L}_{WM} = \mathop{\mathbb{E}}_{\substack{x_s, x_t \sim \text{nonPOI} \\ w \sim \{0,1\}^{n_w}}} \sum_{i=1}^{n_w} \Big( & w_i \log \sigma\left(D_w\left(y_{s,t}\right)\right)_i \\ & + (1 - w_i) \log\left(1 - \sigma\left(D_w\left(y_{s,t}\right)_i\right)\right) \Big), \end{aligned} \quad (7)$$

where $y_{s,t} = G_{SS}(x_s, x_t)$, $D_w$ denotes the watermark decoder, $\sigma(\cdot)$ denotes the sigmoid activation function, $n_w$ is

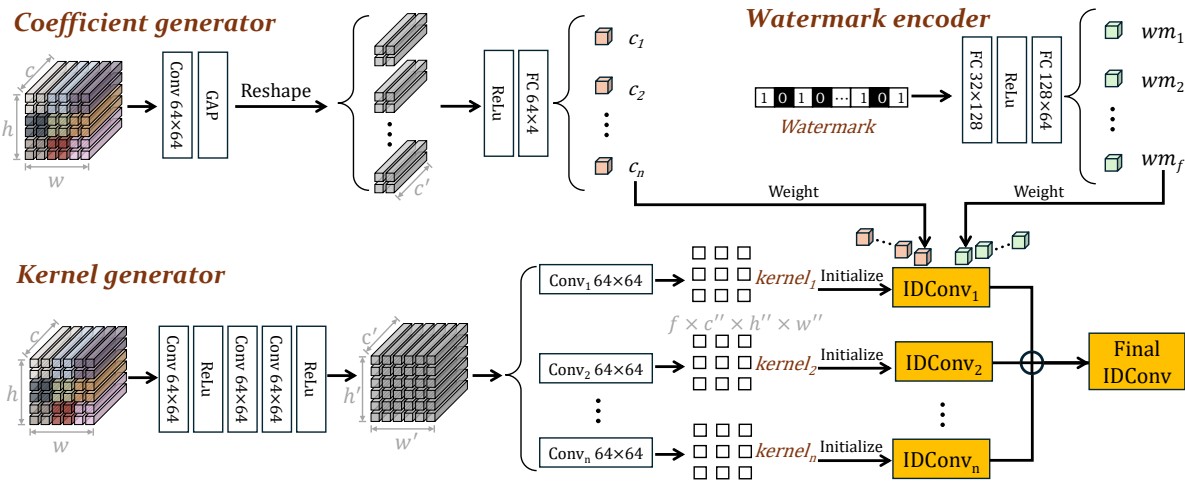

*Figure 4.* Kernel generator (bottom), coefficient generator (top left) and watermark encoder (top right) for IDConv construction. The specific network structure parameters shown in the figure use BlendFace as an example. Different face swap models may have variations in parameters but the overall generation process remains basically the same.

the length of the watermark $w$, and $w_i$ is the $i$-th bit of the watermark. Note that during training, $D_w$ learns to decode different watermarks using randomly generated $w$ as input to the passport layer of the model. Once the model has been trained, a fixed $w$ is selected by the model developer to program the corresponding *WM* into the *IDconv* kernels of the passport layer through the watermark encoder. Thereafter, the watermark encoder is detached from ID passport layer.

The overall loss function used for training the *Secure Swap* model is given by:

$$\mathcal{L} = \lambda_1 \mathcal{L}_{\text{FS}} + \lambda_2 \mathcal{L}_{\text{kd}} + \lambda_3 \mathcal{L}_{\text{POI}} + \lambda_4 \mathcal{L}_{WM}, \qquad (8)$$

where $\lambda_1$ to $\lambda_4$ are the hyper parameters used to regulate the respective loss functions.

## 4. ID Passport Layer

Details of the kernel generator, coefficient generator and watermark encoder involved in the construction of IDConv are shown in Fig. 4. Their purposes are delineated as follows:

**Kernel Generator:** Different input images can yield different weights, reflecting diverse identity-related information. This is leveraged to extract the identity-related latent features from $F$ through a series of convolution layers and ReLU activation layers to produce multiple kernels. These kernels are initialized and used in subsequent IDConv layers.

**Coefficient generator:** When each IDConv inherits identical kernel parameters from $F$, consecutive convolution layers could end up performing the same operation, which could hinder convergence. To prevent this, it is essential to introduce variations among the IDConv blocks by weighing the $n$ kernels with $n$ different coefficients, $c_1, c_2, \cdots, c_n$.

The coefficients are derived from $F$ through a convolution (Conv) layer, an average pooling (Pool) layer and a fully connected (FC) layer, followed by a ReLu and a FC layer after reshaping.

**Watermark Encoder:** It uses randomly generated binary bits during training and the hashed code of a developer's selected message after training to generate a watermark vector *WM* through a cascade of a FC-ReLu-FC layers. The elements, $wm_1, wm_2, \cdots, wm_f$, are converted into weights in the $f$ channel dimensions of each IDConv.

## 5. Experimental Results and Discussions

### 5.1. Datasets and models

We utilize CelebA (Liu et al., 2015), which contains over 10599 identities, serves as nonPOI dataset to fine-tune the face swap model. To prevent the model from learning dataset-specific features instead of POI identity, two distinct datasets VGGFace2 (Cao et al., 2018) and PubFig83 (Kumar et al., 2009) are mixed to serve as POI dataset.

We conduct experiments on various face swap models, including SimSwap (Chen et al., 2020), Faceshifter (Li et al., 2020), BlendFace (Shiohara et al., 2023), and MobileFS-GAN (Yu et al., 2022a). When training these models, we primarily adopt the default parameters explicitly specified in their official implementation. The weights of different loss terms in the total training loss $\lambda_1$, $\lambda_2$, $\lambda_3$ and $\lambda_4$ are all set to 1.0. Other training details are provided in *Appendix*.

### 5.2. Fidelity analysis

Fidelity refers to the faithful reproduction of the original face swap functionality from the face swap model $G$ to

*Table 1.* Comparison of the absolute and relative identity and attribute fidelities. The bracketed figure for each metric refers to their difference.

| Model | Type | Identity Similarity/Attribute Similarity | | |
|---|---|---|---|---|
| | | Arc/Arc-R | Blend/Blend-R | Attr/Attr-R |
| SimSwap | $G$ | 0.148/0.386 | 0.024/0.159 | 0.880/0.572 |
| | $G_{SS}$ | 0.148/0.489 | 0.030/0.229 | 0.866/0.565 |
| | | (+0.000/+0.103) | (+0.005/+0.070) | (-0.014/-0.007) |
| FaceShifter | $G$ | 0.767/0.847 | 0.466/0.851 | 0.831/0.543 |
| | $G_{SS}$ | 0.823/0.897 | 0.496/0.901 | 0.841/0.538 |
| | | (+0.055/+0.050) | (+0.029/+0.049) | (+0.010/-0.005) |
| BlendFace | $G$ | 0.217/0.322 | 0.106/0.409 | 0.954/0.560 |
| | $G_{SS}$ | 0.188/0.324 | 0.056/0.435 | 0.950/0.558 |
| | | (-0.029/+0.002) | (-0.050/+0.026) | (-0.004/-0.002) |
| Mobile FSGAN | $G$ | 0.488/0.537 | 0.367/0.611 | 0.866/0.557 |
| | $G_{SS}$ | 0.523/0.591 | 0.404/0.683 | 0.857/0.553 |
| | | (+0.034/+0.054) | (+0.037/+0.071) | (-0.009/-0.004) |

the secure model $G_{SS}$. Here, fidelity can be quantitatively assessed on two criteria: (1) the success rate of face swap for nonPOI, i.e., whether the identity of target image has been correctly altered to that of the source image; (2) the visual quality of the swapped images for nonPOI.

*Table 2.* Perceptual quality comparison of nonPOI face swapped images between $G_{SS}$ and $G$ across varying numbers of POI IDs.

| Models | Number of POI | PSNR ↑ | SSIM ↑ | LPIPS ↓ |
|---|---|---|---|---|
| BlendFace | 128 | 34.342 | 0.982 | 0.018 |
| | 512 | 34.359 | 0.981 | 0.016 |
| | 1024 | 34.157 | 0.976 | 0.010 |
| SimSwap | 128 | 33.965 | 0.965 | 0.034 |
| | 512 | 34.854 | 0.964 | 0.062 |
| | 1024 | 34.604 | 0.951 | 0.044 |
| FaceShifter | 128 | 34.602 | 0.968 | 0.041 |
| | 512 | 34.168 | 0.968 | 0.031 |
| | 1024 | 33.915 | 0.965 | 0.033 |
| Mobile FSGAN | 128 | 34.015 | 0.952 | 0.024 |
| | 512 | 33.390 | 0.957 | 0.023 |
| | 1024 | 33.413 | 0.959 | 0.021 |

To evaluate the first criterion, we analyze the face swap outputs of $G$ and $G_{SS}$ for the same pair of input images by comparing their absolute and relative similarity in terms of identity and attributes.To increase the confidence of our evaluation, we use two different encoders, *ArcFace (Arc)* (Deng et al., 2019) and *BlendFace (Blend)* (Shiohara et al., 2023), as $E_{id}$ for the identity embedding. The calculation methods for absolute/relative identity similarity and attribute similarity are provided in the *Appendix*. A high attribute similarity indicates that only the identity has been swapped while everything else remains unchanged. A high identity similarity means that the swapped face's identity is similar to the expected identity. What we need to focus on is the difference in score between $G$ and $G_{SS}$ on these metrics. The smaller the difference, the better the fidelity of $G_{SS}$. As in Table 1, the changes in the identity distance (*Arc*, *Arc-R*, *Blend*, *Blend-R*) and attribute distance (*Attr*, *Attr-R*) metrics

between $G$ and $G_{SS}$ are generally very small, often within 0.05 for most models. This indicates that the success rate of face swapping for nonPOI images is preserved.

To evaluate the second criterion, we evaluate the perceptual differences between images generated by $G$ and $G_{SS}$ with Peak Signal-to-Noise Ratio (PSNR) (Hore & Ziou, 2010), Structural Similarity Index Measure (SSIM) (Hore & Ziou, 2010) and LPIPS (Zhang et al., 2018). Higher PSNR values indicate better similarity between two images. SSIM ranges from 0 to 1, with values closer to 1 representing higher similarity. LPIPS quantifies perceptual differences, where lower values indicate smaller differences. As shown in Table 2, we observed that with POI numbers varying from 128 to 1024, $G_{SS}$ consistently upholds the visual quality of swapped face images, with PSNR above 32, SSIM exceeds 0.95, and LPIPS averages below 0.05. This implies that the visual quality of nonPOI is preserved.

### 5.3. Effectiveness analysis

Effectiveness is verified by the success rate of POI redaction, and the watermark performance on nonPOI outputs. We adopt $SR_{mask}$ from (Huang et al., 2022) for the measurement of the success rate of POI redaction. We also utilize *MSE*, $ACC_{WM}$ and *TPR* to evaluate watermark performance. The calculation details are shown in the *Appendix*. A higher $SR_{mask}$, $ACC_{WM}$ and *TPR* score indicates better effectiveness while a lower *MSE* indicates better visual quality. Table 3 shows the $SR_{mask}$, $Acc_{WM}$, and TPR of various Secure Swap models with different number of POI IDs. In most cases, the average watermark extraction accuracy exceeds 99%. The exception is MobileFSGAN, which has a slightly lower $Acc_{WM}$ of around 97%. Considering the high TPR, 97% accuracy is not an obstacle to effective provenance tracking.

*Table 3.* Effectiveness of POI redaction and watermark.

| Models | POI IDs | $SR_{mask}$ ↑ | MSE↓ | $Acc_{WM}$ ↑ | TPR↑ |
|---|---|---|---|---|---|
| BlendFace | 128 | 1.000 | 0.000 | 0.987 | 0.9999 |
| | 512 | 1.000 | 0.000 | 0.999 | 0.9999 |
| | 1024 | 1.000 | 0.000 | 0.995 | 0.9999 |
| SimSwap | 128 | 1.000 | 0.000 | 0.998 | 0.9999 |
| | 512 | 1.000 | 0.000 | 1.000 | 0.9999 |
| | 1024 | 1.000 | 0.000 | 0.999 | 0.9999 |
| FaceShifter | 128 | 1.000 | 0.014 | 0.990 | 0.9999 |
| | 512 | 1.000 | 0.051 | 0.999 | 0.9999 |
| | 1024 | 1.000 | 0.021 | 1.000 | 1.0000 |
| Mobile FSGAN | 128 | 1.000 | 0.000 | 0.985 | 0.9998 |
| | 512 | 1.000 | 0.184 | 0.976 | 0.9990 |
| | 1024 | 1.000 | 0.000 | 0.982 | 0.9997 |

To further assess how our watermark embedding influences the model's generative performance, we conducted a com-

parison with the latest watermarking approaches such as arrtributing (Nie et al., 2023), Stable Signature (Fernandez et al., 2023), PluggableWM (Bao et al., 2024), WFWM (Fei et al., 2024). Higher PSNR and SSIM values indicate better watermark imperceptibility. As shown in Table 4 on Page 7 left column, our method shows excellent performance in imperceptibility.

*Table 4.* Performance comparison of different watermark methods. The model used for comparison is BlendFace.

| Watermark works | PSNR ↑ | SSIM ↑ |
|---|---|---|
| Arrtributing (ICML2023) | 28.149 | 0.903 |
| Stable Signature (ICCV2023) | 29.250 | 0.911 |
| PluggableWM (TDSC2024) | 31.284 | 0.939 |
| WFWM(TIFS2024) | 34.165 | 0.949 |
| **Ours** | **34.321** | **0.981** |

### 5.4. Robustness analysis

We consider two scenarios of model application: the first involves interactive online use via an API service, which allows applications to communicate with the model over the internet, and the second involves local inference by downloading the model with Secure Swap, as released by the developers, which has the flexibility to run the model in an offline environment. We perform robustness testing for both scenarios to evaluate the model's performance under different deployment conditions.

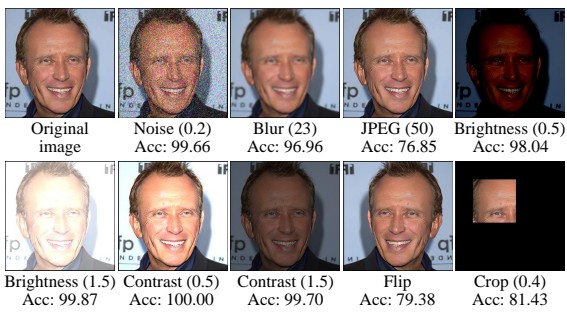

*Figure 5.* NonPOI results under various image-level attacks.

**Image-level Attacks.** In the scenario where users interact with the system via an API service, their only means of communication with $G_{SS}$ is through the input of images, which may expose $G_{SS}$ to various image processing attacks. These include unintentional image degradation during transmission, such as compression artifacts introduced by formats like JPEG, as well as deliberate manipulations, such as blurring, noise addition, or brightness and contrast adjustment. We provide nonPOI results under various image processing attacks on Secure Swap BlendFace model in Fig. 5 as exemplification. Despite significant perturbations such as Noise ($\sigma_{noise} = 0.2$) and Blur (kernal size = 23), the model

maintains very high accuracies of 99.66% and 96.96%, respectively. JPEG (quality factor = 50) compression reduces the accuracy to 76.85%.

Fig. 5 shows strong robustness to Brightness and Contrast adjustments, with accuracy consistently above 98%, while more challenging attacks like Flip and Crop (crop rate = 0.4) result in lower accuracies of 79.38% and 81.43%, respectively.

To enhance the model's robustness, we retrain $G_{SS}$ by randomly applying one of the seven image processing attacks to the nonPOI results for each batch of training. The attacked output images were passed through the watermark decoder, which learns to decode the predefined watermark from these additional attacked samples with the watermark loss. We called the retrained Secure Swap model with enhanced watermark decoding capability the augmented model. The watermark extraction accuracy and PSNR values of the original and augmented Secure Swap BlendFace models are plotted against attack strengths for eight image processing attacks in Fig. 6.

Figure 6 compares the watermark accuracy and PSNR of the original and augmented Secure Swap BlendFace models under varying attack strengths for eight image processing methods. It shows that the augmented model $G_{SS}$ (SS - with aug) consistently outperforms the non-augmented model $G$ (SS - w/o aug) across various image attacks. For instance, in the JPEG Compression and Brightness Enhancement plots, the $Acc_{WM}$ of each augmented model remains high as attack intensity increases, while the non-augmented model's accuracy drops sharply. Under Gaussian Noise and Cropping attack, the augmented model demonstrates a more gradual decline in $Acc_{WM}$ compared to the steeper drop of the non-augmented model. These results indicate that the augmented Secure Swap method can effectively resist strong image processing attacks that degrade image quality significantly.

**Model-level Attacks.** When $G_{SS}$ is downloaded by users for local inference, it becomes potentially vulnerable to white-box model modification attacks. In this scenario, attackers have full access to the model's architecture and parameters, allowing them to thoroughly analyze and manipulate it. Once they become aware of the secure swap mechanism, they may specifically target the ID passport layer. Possible attacks could involve removing the ID passport layer entirely or introducing perturbations to compromise its integrity. Such modifications could undermine the credibility of the model's watermark, leading to unauthorized or malicious use of the model. We evaluate the robustness of $G_{SS}$ against three types of **white-box model attacks**. They are *model quantization, model pruning, and noise addition* to model weights, and *a mixture of these three attacks*. The evaluation results with varying attack strength are shown in

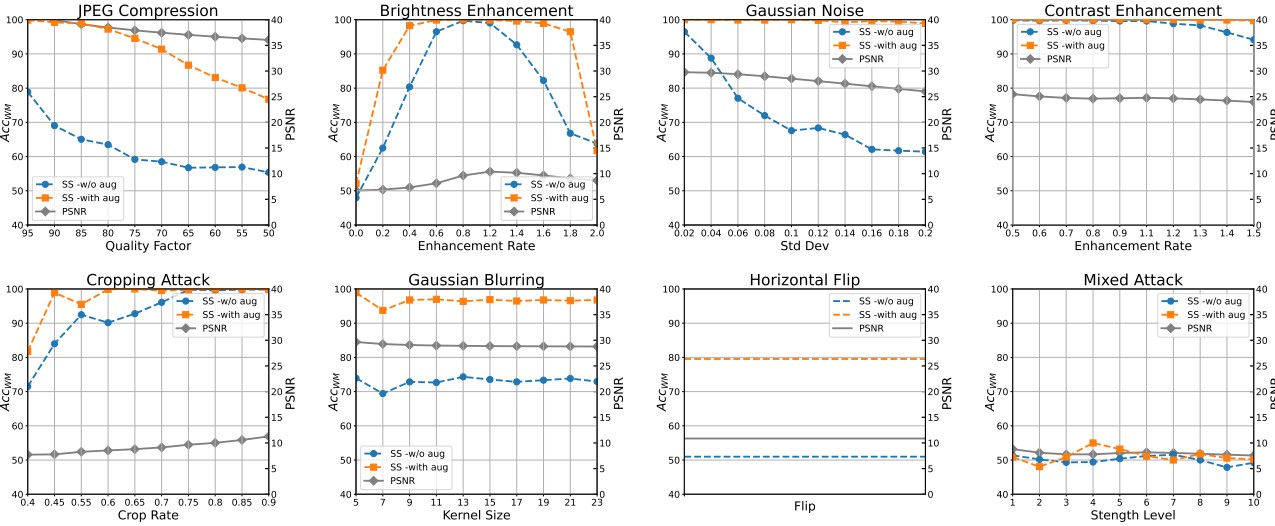

*Figure 6.* Robustness of original and augmented Secure Swap BlendFace model against different strengths of eight different image processing attacks. The grey line indicates average PSNR of output images after the attack.

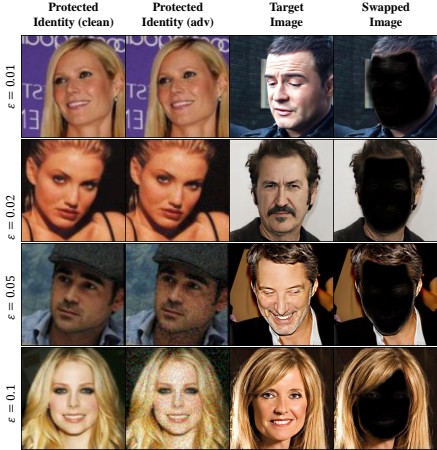

*Figure 7.* POI results of $G_{SS}$ against white-box FGSM.

Fig. 9 of *Appendix*. As attack intensity increases, the PSNR of images from all four attacked models drops significantly, often below 20, and even below 10 under mixed attacks. In contrast, watermark extraction accuracy decreases much more slowly, with $Acc_{WM}$ remaining above 75%—a reasonable verification threshold—even when image quality is severely degraded. This suggests that even with white-box access, attackers cannot remove the watermark without compromising the model's utility.

**Adversarial Attacks.** Adversarial attacks, implemented by adding adversarial noise to images, is a different type of image-level attack (Akhtar & Mian, 2018). It presents a severe threats to the identity of protected individuals if an attacker can use an identity extractor to perform adversarial attacks that successfully prevent SS from "recognizing" the POI. To evaluate this feasibility, we consider the most widely used fast gradient sign method (FSGM) (Dong et al., 2018) and projected gradient descent (PGD) (Madry et al., 2018) adversarial attacks. We directly used the identity extractors embedded within each face swap model, such as ArcFace or BlendFace, to perform these adversarial attacks. Table 5 shows the $SR_{mask}$ of $G_{SS}$ obtained upon the attacks. For both FGSM and PGD, even when the perturbation amplification factor $\epsilon$ has increased to 0.1, $G_{SS}$ can still maintain a $SR_{mask}$ of 100% for POI redation. Fig. 7 displays the outputs of four POI after the FGSM adversarial attack. The first column shows the clean, protected images, which are the photos of the POI. The second column displays the POI images with the adversarial perturbations added. The third column displays the target photos, and the fourth column shows the face swap results. Each row displays the results with different perturbation strengths of FGSM. The adversarial attacks may result in outputs that are not totally black, but the faces are still heavily redacted, and the POI cannot be identified even when the adversarial perturbations are very high and visually perceivable.

### 5.5. Ablation study

**Capacity of POI.** To evaluate this, we increased the number of POI ID to 1024, 2048 and 4096, and measure the protection performance of POI by $SR_{mask}$ and the quality of generated face swapped images for nonPOIs by LPIPS. The results are shown in Table 6, where the values of $SR_{mask}$ and LPIPS are given before and after the delimiter '/', respectively. It is observed that the LPIPS scores for all Secure Swap models are very low and remain almost unchanged across all three numbers of POI IDs, indicating that their generated images are close to real images in both quality and diversity, and they can accommodate a large number of POI IDs without compromising the generated image qual-

ity. Moreover, the success rates of POI redation are 100% throughout all evaluations. Combining with the LPIPS results, it can be concluded that $G_{SS}$ can effectively protect at least 4K+ individuals as POI without compromising the model performance on nonPOI images.

*Table 5.* $SR_{mask}$ of different $G_{SS}$ upon FGSM and PGD attacks with different perturbation strengths ($\epsilon = 0.01$ to $0.1$).

| Models | Attack | 0.01 | 0.02 | 0.05 | 0.1 |
|---|---|---|---|---|---|
| BlendFace | FGSM | 1.00 | 1.00 | 1.00 | 1.00 |
| | PGD | 1.00 | 1.00 | 1.00 | 1.00 |
| SimSwap | FGSM | 1.00 | 1.00 | 1.00 | 1.00 |
| | PGD | 1.00 | 1.00 | 1.00 | 1.00 |
| FaceShifter | FGSM | 1.00 | 1.00 | 1.00 | 1.00 |
| | PGD | 1.00 | 1.00 | 1.00 | 1.00 |
| MobileFSGAN | FGSM | 1.00 | 1.00 | 1.00 | 1.00 |
| | PGD | 1.00 | 1.00 | 1.00 | 1.00 |

*Table 6.* $SR_{mask}$ / $LPIPS$ of different face swap models with different number of protected identities.

| Models / Identities | 1024 | 2048 | 4096 |
|---|---|---|---|
| BlendFace | 1.000 / 0.018 | 1.000 / 0.018 | 1.000 / 0.018 |
| SimSwap | 1.000 / 0.060 | 1.000 / 0.061 | 1.000 / 0.063 |
| FaceShifter | 1.000 / 0.043 | 1.000 / 0.043 | 1.000 / 0.044 |
| MobileFSGAN | 1.000 / 0.021 | 1.000 / 0.022 | 1.000 / 0.024 |

**Training Dataset Requirement.** For some POI, the availability of high-quality images may be limited. To study the impact of training dataset size on success rate of POI redation, we trained the SS model with different sizes of POI training sets, each training set contains different numbers of images per POI, and tested the model performance on unseen POI images. The results are shown in Table 7, it shows that the Secure Swap models perform well in POI redation even when they are trained with relatively small datasets. For any POI, $SR_{mask}$ of more than 95% can be achieved with just 4 images, and increasing the dataset size to 16 images yields 100% success rate of POI redation.

*Table 7.* $SR_{mask}$ of different face swap models with different numbers of training photos of protected identity.

| Models \Photos | 1 | 2 | 4 | 8 | 16 |
|---|---|---|---|---|---|
| BlendFace | 0.965 | 0.982 | 0.990 | 0.994 | 1.000 |
| SimSwap | 0.942 | 0.962 | 0.987 | 0.989 | 1.000 |
| FaceShifter | 0.907 | 0.954 | 0.973 | 0.987 | 1.000 |
| MobileFSGAN | 0.915 | 0.931 | 0.970 | 0.994 | 1.000 |

**Training Overhead.** Table 8 presents the runtime required for training $G$ and $G_{SS}$ on a workstation equipped with one NVIDIA A100 GPU (80 GB), an Intel Xeon Platinum 8358 CPU@2.60 GHz, and 1024 GB of memory. It shows that compared to $G$, $G_{SS}$ requiring slightly more time per iteration but significantly less time for full training.

*Table 8.* Comparison of Runtime for Each Iteration and Complete Training of Models $G$ and $G_{SS}$

| Model | Running Time (G / $G_{SS}$) | |
|---|---|---|
| | Iteration | Training |
| BlendFace | 1.15 / 1.41s | 95h 50m / 7h 52m |
| SimSwap | 0.67 / 0.91s | 93h 3m / 5h 4m |
| FaceShifter | 0.96 / 1.19s | 132h 58m / 6h 40m |
| MobileFSGAN | 0.72 / 0.99s | 48h 16m / 5h 47m |

**Limitation and exploration: quantitative boundaries for newly added POI.** In our experimental setup, POI number is approximately 1,000. When new POI is added, the model requires retraining to accommodate the updated dataset. However, as shown in Table 8, our method features a low training overhead, making the retraining process significantly less time-consuming. Furthermore, to explore the quantitative boundaries for newly added POI while ensuring watermark performance, we conduct experiments by adding POI numbers from 1K to 10K. As shown in Table 9, the gradual increase of POI number does not significantly impact the model's ability to skip irrelevant POIs or maintain watermarking effectiveness.

*Table 9.* Performance of POI redaction and watermark embedding when POI number arising

| POI number | 1k | 3k | 5k | ~10k |
|---|---|---|---|---|
| $SR_{mask}$ ↑ | 1.000 | 1.000 | 0.995 | 0.989 |
| $FID$ ↓ | 2.39 | 2.33 | 2.26 | 2.18 |
| $PSNR$ ↑ | 34.321 | 34.402 | 34.026 | 33.897 |

# 6. Conclusion

Our work proposed a robust Secure Swap mechanism to protect POI from unauthorized face swaps and embed watermarks in the generated images of nonPOI to facilitate provenance tracking. We designed an ID Passport layer to generate unique identity-specific feature maps from diverse upstream latent features of each pair of source and target images. Watermark is also embedded into nonPOI output images through the ID Passport layer by training a detachable watermark encoder to transform randomly generated watermarks into weights of the ID passport layer, and a corresponding decoder to extract these watermarks from the generated face swapped outputs. The trained encoder can be attached to any instances of the trained Secure Swap model to embed customer-specific watermark for provenance tracing. Notably, our method exhibits outstanding robustness against image-level, model-level, and adversarial attacks. In the future, we aim to adapt Secure Swap to emerging diffusion-based generative models to enhance security and promote ethical, accountable applications of generative AI in legitimate face swap technologies.

## Acknowledgment

This work was supported in part by the National Natural Science Foundation of China under Grants 62472454 and U2336208. This research was also supported by the Ministry of Education, Singapore, under its Academic Research Fund (AcRF) Tier 2 Award No. MOET2EP50220-0003. The authors would like to express their sincere gratitude for the generous support provided by these funding agencies.

## Impact Statement

This work contributes to the development of secure and accountable generative AI systems by proposing Secure Swap, a robust face swapping framework that integrates identity protection and provenance tracking. The introduction of the ID Passport layer and detachable watermarking mechanism enables scalable deployment and individualized traceability. Our method enhances the reliability of face synthesis technologies under real-world attack scenarios. Looking forward, adapting this framework to diffusion-based models aligns with the industry's move toward more secure, ethical, and controllable generative applications, supporting responsible innovation in AI-driven media generation.

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

# A. Responsible Face Swapping: Governance and Security

In this section, we first outline the specific scenario and problems to which our method applies under the foundation for responsible AI governance. To set the stage for the two security problems addressed by our solution, we then introduce the major stakeholders in this framework and their roles, incentives, privileges and constraints.

## A.1. Responsible face swapping governance

Our work aims to provide a feasible and efficient solution for model developers to responsibly release their face swap models under the responsible generative AI governance framework. Three main parties are involved in this ideation. **Developers:** This role possesses full control of the model architecture, training strategy, and dataset (including the selection of POI). Developers release online API or open-source models to provide face editing services. Note that the training method used for the traceable watermarking component is proprietary to the developers and will not be publicly released for security reasons. **Authority:** This role possesses the right to require developers to fully disclose watermarking information and keep records of watermarking traceability when publishing models. They also have the right to investigate and collect evidence on the use of non-compliant models, and then penalize the subjects of the offense. **Users:** *(a) regular users*, this role only utilizes the provided service to swap ordinary faces that do not involve any POI. Their swapped images are used solely for legitimate purposes. *(b) malicious users*, this role has two intents. One intent is to directly forge POI images through the provided face swapping service. Upon finding that the POI swap failed, they may try to undermine the POI protective functionality of the model when downloaded it for local inference. The other intent is to directly forge nonPOI images for fraud. Upon finding that the swapped images carrying watermarks, they may try to remove the watermark or undermine the watermark generation mechanism. Their presence is the very reason for the problems tackled in this work.

## A.2. Celebrity protection and provenance tracing

We tackle the security of responsible face swapping on two fronts. Firstly, due to the incredible spotlight on celebrities, we aim to provide security to celebrities (POI) in face swap models to prevent their facial image from being abused. Secondly, we aim to make each developer's model identifiable from its generative outputs for forensic analysis to track the authenticity and source of generative images of ordinary individuals (nonPOI). Under the responsible generative AI governance framework, developers are advocated to institute transferable forensic features into their model building

process to facilitate identification of generative images from real images and traceability of downstream distribution to allow the authority to enforce ethical and responsible use of generative model at a copyright or crime tribunal.

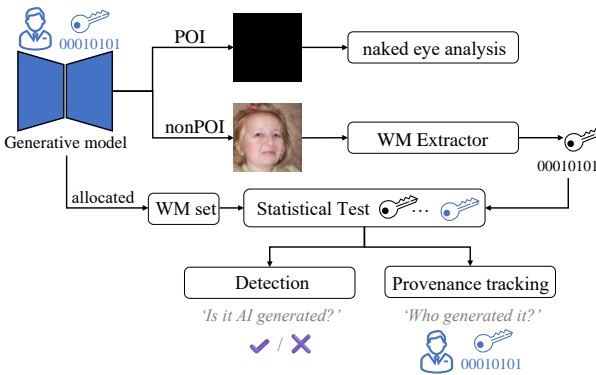

*Figure 8.* The process of image watermark (WM) detection and provenance tracking after deploying **Secure Swap**.

Fig. 8 epitomizes the essence of our proposition towards achieving celebrity protection and provenance tracing. Given the many-to-one relationship between faces and identity, model modification centered on identity offers a more effective approach to ensure the model responds with differentiated behaviors towards different identities. A model developer can forbid POI swap attempt by preemptively forcing its model to redact POI face from the output in a way that can be easily detected by naked eye. For the swapped output of nonPOI inputs, two questions are to be addressed in this work. **1. *Detection***: 'Is this image generated by AI?', and **2. *Provenance tracking***: 'Who generated this image?'.

In the era of generative AI, criminal justice system is increasingly challenged by the indistinguishability between "original" content produced by generative models and real content. Watermarking is a well-accepted technology to proactively mark content to convey authenticity. Tech companies like Google, Microsoft, Meta, and Amazon have all pledged to develop watermarking systems for identifying AI-generated content. Typically, each developer can have a unique predefined watermark registered with the authority. The developer's model can learn to embed this watermark invisibly into the generated images of nonPOI inputs. The watermark embedded in the swapped outputs can be decoded by a watermark extractor for statistical analysis. If a watermark is detected in a suspect image, it can be determined that it is a forged image generated by a face swap model. If the detected watermark matches the watermark of a specific model, it can be inferred that the image was generated by that particular face swap model.

Unlike the methods discussed in Sec. 2, we aim to empower model developers to embed proactive security features di-

rectly within face swap models. This will allow the models to preemptively block any malicious attempts targeting specific identities while also watermarking the generated content. By addressing potential threats at the source, it prevents the creation of fake images without requiring any preprocessing of input images. Furthermore, we would like to safeguard *all images* containing any protected identities, rather than selectively targeting certain images. This is to ensure that the model consistently applies security measures across all relevant outputs. For practical implementation and cost efficiency, it is also important to reduce the development expenses while maintaining high levels of security and reliability by training the model once but the outputs of multiple instances of the same model can still be embedded with their respective unique instance watermarks for provenance tracking.

**Watermark detection.** Each instance (copy) of a trained secure swap model $G_{SS}$ is watermarked before it is released. To watermark an instance, a unique copyright or license message selected by the developer is first hashed into a binary watermark $w$ of length $n_w$. Then the trained watermark encoder is attached to the ID passport layer of the instance to program $w$ into its *IDconv* kernels. The watermark encoder is then detached before the watermarked instance is released. To detect if the watermark $w \in \{0,1\}^{n_w}$ programmed into a distributed instance $G_A$ is present in a nonPOI image $x$, we can pass $x$ to the trained watermark decoder $D_w$ to extract the watermark $w'$. The detection test (Fernandez et al., 2023; Yu et al., 2021) checks the number of matching bits, $Match(w, w')$, between $w$ and $w'$ against a non-negative threshold $\tau \leq n_w$. If $Match(w, w') \geq \tau$, the image $x$ is highly probable to be generated from $G_A$.

To increase the confidence of watermark detection, we perform a hypothesis test on $\tau$. We test the statistics of the alternative hypothesis $H_1$: *"x was generated by a model $G_A$ instance"* against the null hypothesis $H_0$: *"x was not generated by model $G_A$"*. Under $H_0$, we assume that all bits $w_i' \ \forall i \in [1, n_w]$ are independent and identically distributed (i.i.d.) Bernoulli random variables with parameter 0.5. Then $Match(w, w')$ follows a binomial distribution with parameters $(n_w, 0.5)$. This assumption has been experimentally verified by (Fernandez et al., 2023). Noted that if a non-watermarked image yields a watermark matching score greater than the threshold $\tau$, the image is considered a false positive sample. The False Positive Rate (FPR) is the probability that $Match(w, w')$ takes a value bigger than the threshold $\tau$. It is obtained from the Cumulative Distribution Function (CDF) of the binomial distribution, and a closed-form expression of FPR can be expressed with the regularized incomplete beta function $I_z(a, b)$ as follows:

$$\text{FPR}(\tau) = P(Match(w, w') > \tau | H_0) = I_{1/2}(\tau+1, n_w-\tau).$$
(9)

The True Positive Rate (TPR) can be evaluated as follows:

$$\text{TPR}(\tau) = \binom{n_w}{\tau} p_w^\tau (1 - p_w)^{n_w - \tau},$$
(10)

where $p_w$ represents the experimental watermark detection accuracy. Using (9) and (10), we can balance the FPR and TPR to achieve a more accurate watermark detection.

**Watermark for provenance tracking.** Each watermark $w$ hashed from the developer's selected copyright message can be assumed to be a i.i.d binary string drawn from the set $\{0,1\}^{n_w}$ for embedding into a model distributed to a downstream commercial customer. Let $w[i]$ denotes the watermark embedded into the model $G_i$ distributed to the customer $D_i$, where $i = 1, 2, \cdots, N$ and $N$ is the total number of customers. If a model is abused for generating a face swapped image $x$ for fraud, the specific downstream commercial customer $D_i$ can be reliably traced by extracting the watermark $w'$ from the face swap image $x$ to comparing against $\{w[1], w[2], \cdots, w[N]\}$.

For provenance tracking, we have $N$ detection hypotheses. If all $N$ hypotheses are disproved, we infer that $x$ is not generated by any of the models distributed to the downstream commercial customers. Otherwise, we attribute the image $x$ to a specific model $G_j$ of the downstream commercial customer $D_j$, where

$$j = \underset{i=1,2,\cdots,N}{\arg\max} \big\{ Match(w', w[i]) \big\}.$$
(11)

Given that $N$ tests are being conducted, the probability of encountering false positives in this detection process is elevated. The global FPR at a given threshold $\tau$ is:

$$\text{FPR}(\tau, N) = 1 - (1 - \text{FPR}(\tau))^N \approx N \cdot \text{FPR}(\tau). \quad (12)$$

Through careful management of detection thresholds and multiple hypothesis testings, reliable provenance tracking can be achieved.

## B. Training details.

When training these models, we primarily adopt the default parameters explicitly specified by the authors in their official implementation. For SimSwap (Chen et al., 2020), we train the model with batch size 32 for 500,000 steps using the Adam optimizer with a learning rate of 0.0004, beta 1 is 0.0, and beta 2 is 0.999. For Faceshifter (Li et al., 2020), we train the model with batch size 16 for 500,000 steps using the Adam optimizer with a learning rate of 0.0004, beta 1 is 0.0, and beta 2 is 0.999. For BlendFace (Shiohara et al., 2023), we train the model with batch size 16 for 500,000 steps using the Adam optimizer with a learning rate of 0.0001, beta 1 is 0.5, and beta 2 is 0.999. The discriminator is updated five

times for each generator update. For MobileFSGAN (Yu et al., 2022a), we train the model with batch size 64 for 500,000 steps using the Adam optimizer with a learning rate of 0.0002, beta 1 is 0.5, and beta 2 is 0.999. The discriminator is updated twice for each generator update. The learning rate for models are linearly decayed to zero from 250,000 steps.

## C. Fidelity analysis

The identity distance is calculated by the cosine similarity between the normalized face embeddings of the source and swapped images (Shiohara et al., 2023). A higher distance implies a successful face swap. To increase the confidence of our evaluation, we use two different encoders, *ArcFace (Arc)* (Deng et al., 2019) and *BlendFace (Blend)* (Shiohara et al., 2023), as $E_{id}$ for the identity embedding. Apart from identity distance, we also utilize *Attribute (Attr)*[2], an attribute extractor, to calculate the attribution distances between the source and swapped images. A smaller attribute distance indicates a better face swap performance. To eliminate the influence of irrelevant parameters such as vector size and vector dimension, we calculate the cosine similarity of extracted feature vectors for *ArcFace*, *BlendFace* and *Attribute*. Besides similarity in identity between the source and swapped images, it is crucial that the swapped images are dissimilar to target images visually. For a more comprehensive evaluation, we also calculate the relative identity and attribute similarity that account for both the source and target images, which are annotated by "-R", as follows (Kim et al., 2022):

$$\mathcal{S}\text{-}R = \frac{\mathcal{S}(x_s, G_{SS}(x_s, x_t))}{\mathcal{S}(x_s, G_{SS}(x_s, x_t)) + \mathcal{S}(x_t, G_{SS}(x_s, x_t))}, \quad (13)$$

where $\mathcal{S}$ here refers to the similarity of identity or attribute between two face images with range $[-1, 1]$.

## D. Effectiveness analysis

$SR_{\text{mask}}$ can be calculated by:

$$SR_{mask} = \frac{1}{n_{ID}} \sum_{i=1}^{n_{ID}} \mathbb{1}(L_2(G, G_{SS}) > 0.05), \quad (14)$$

where $n_{ID}$ denotes the number of POI IDs. $\mathbb{1}(\cdot)$ is an indicator function, which produces 1 if the condition in the argument is true and 0 otherwise. $L_2(G, G_{SS})$ denotes the weighted $L_2$ norm between the outputs of $G$ and $G_{SS}$. Following (Huang et al., 2022), we consider the POI redation to be successful if $L_2(G, G_{SS}) > 0.05$. The weighted $L_2$ norm in (14) can be determined by the pixel-level discrepancies between $G$ and $G_{SS}$ when fed with POI images as

---

[2]https://github.com/Hawaii0821/FaceAttr-Analysis

follows:

$$L_2(G, G_{SS}) = \frac{\sum_i \sum_j \mathcal{M}_{(i,j)} \left\| G(x)_{(i,j)} - G_{SS}(x)_{(i,j)} \right\|}{\sum_i \sum_j \mathcal{M}_{(i,j)}}, \quad (15)$$

where $x = (x_s, x_t)$ are the input faces, and $(i, j)$ is the pixel coordinate of $x$. The binary mask $\mathcal{M}_{(i,j)}$ is determined by the pixel discrepancy between the edited image $G(x)$ and the original image $x$ at $(i, j)$ in (16).

$$\mathcal{M}_{(i,j)} = \begin{cases} 1, & \text{if } \left\| G(x_s, x_t)_{(i,j)} - (x_s)_{(i,j)} \right\| > 0.5 \\ 0, & \text{otherwise} \end{cases}, \quad (16)$$

## E. Model-level attack details

We evaluate the robustness of $G_{SS}$ against three types of **white-box model attacks**. They are *model quantization, model pruning, and noise addition* to model weights, and *a mixture of these three attacks*. We also vary the attack strength of each type of attack. For noise addition, we added Gaussian noise with varying standard deviations (Std Dev or $\sigma_{noise}$). For pruning, we set a certain ratio of weights to zero based on their absolute values. For quantization, we set the weight precision to decimal numbers with varying numbers of fractional digits. The evaluation results with varying attack strength are shown in Fig. 9.

## F. Analysis of loss functions.

We balance multiple losses by assigning equal weights to each term. This is intentional and based on the following reasons: The watermark encoder, kernel generator, and coefficient generator have their learnable parameters, which enable our losses to achieve stable convergence without requiring careful weighting. Besides, all components of the model are optimized jointly and simultaneously. The overall loss function consists of face-swapping losses (FS), a knowledge distillation loss (KD), a POI-specific loss, and a WM loss. FS and KD are applied to nonPOI samples, while the POI and WM losses act on ID protection and watermark embedding, respectively, which are non-conflicting. Equal weights are assigned to all components based on three observations: (I) Numerical scales remain comparable across losses; (II) FS and KD address similar objectives, and all losses operate on disjoint data partitions (nonPOI vs. POI); (III) The WM loss modifies only imperceptible features, preserving image quality and avoiding interference with other losses.

We also acknowledge the importance of each loss component introduced to address a specific task: (I) FS loss and KD loss: enhances visual fidelity in face-swapped images. (II) POI loss: enforces protection of POI identities. (III)

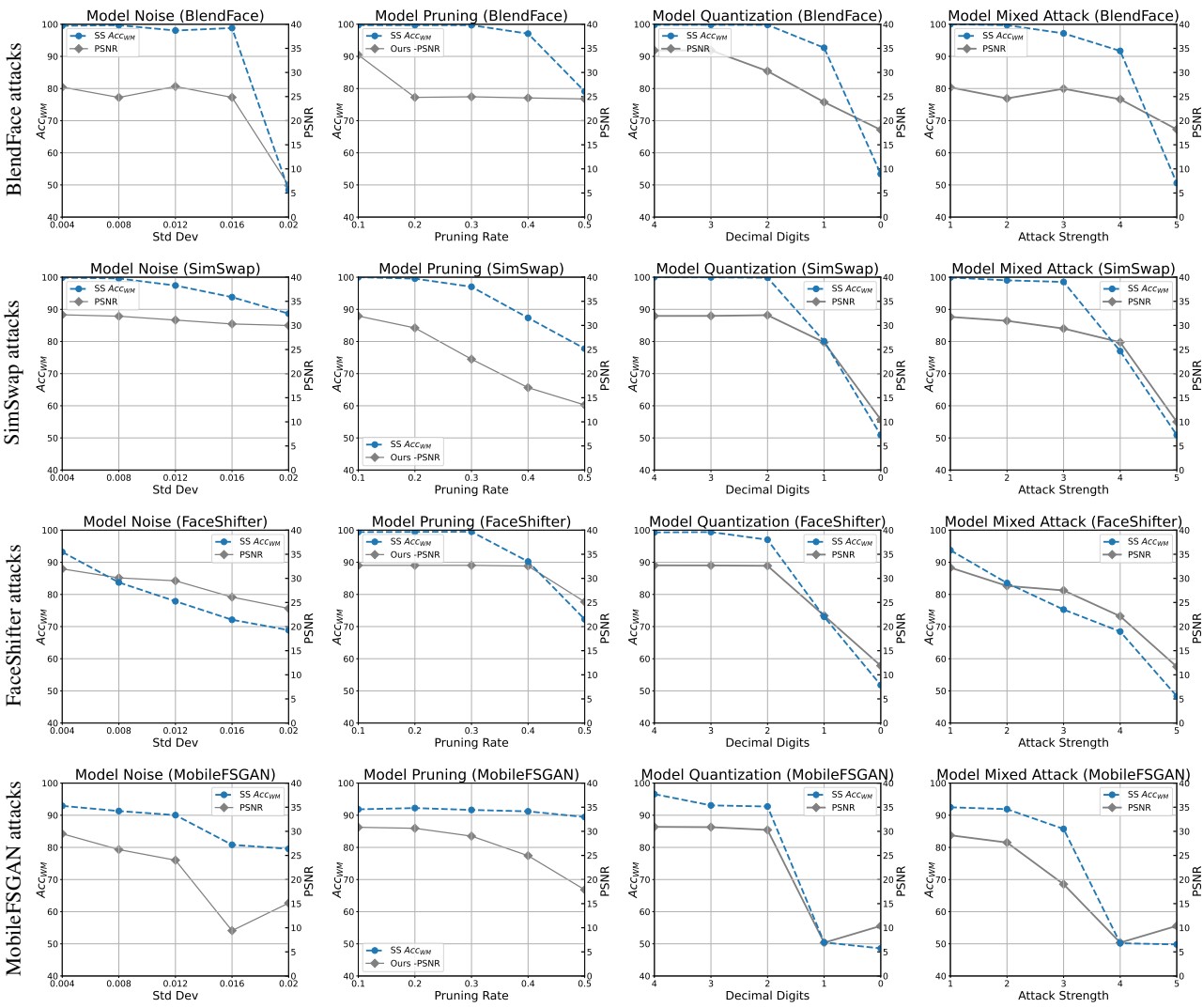

*Figure 9.* Watermark accuracy ($Acc_{WM}$) of Secure Swap (a) BlendFace (b) SimSwap (c) FaceShifter and (d) MobileFSGAN models after attacks of model noising (1st column), pruning (2nd column), quantization (3rd column) and mixed attack (4th column).

WM loss: embeds watermark into the output of nonPOI images.

To evaluate their contributions, we took BlendFace as an example and conducted ablation studies by removing each loss independently in the table below. The results show significant performance degradation in the corresponding tasks on the removal of any loss term.

*Table 10.* Performance comparison of different loss configurations.

| Loss | PSNR | SRmask | WM_Acc |
|------|------|--------|--------|
| w/o FS loss | 32.3 | 0.93 | 0.98 |
| w/o KD loss | 27.7 | 1.00 | 0.99 |
| w/o POI loss | 34.5 | 0.00 | 0.99 |
| w/o WM loss | 35.2 | 1.00 | 0.51 |

## G. Stability testing of our watermark performance

Our watermarking method demonstrates promising performance in both effectiveness and usability. In terms of efficiency, by binding the watermark to model weights, our method enables scalable watermarking for efficient deployment in distribution scenarios. To address the potential influence of randomness, we report the average accuracy and standard deviation over 1,000 different embedded watermarks, as shown in the Table 11. The low standard deviations indicate consistent performance across watermark instances. These results suggest that our method remains robust and is not affected by randomness.

*Table 11.* Performance metrics of different models.

| Model | BlendFace | SimSwap | FaceShifter | MobileFSGAN |
|-------|-----------|---------|-------------|-------------|
| Acc | 99.48 | 99.93 | 100.0 | 98.24 |
| std | 0.33 | 0.29 | 0.00 | 0.30 |

## H. Training Overhead.

Table 8 presents the runtime required for training $G$ and $G_{SS}$ on a workstation equipped with one NVIDIA A100 GPU (80 GB), an Intel Xeon Platinum 8358 CPU@2.60 GHz, and 1024 GB of memory. It shows that compared to $G$, $G_{SS}$ requiring slightly more time per iteration but significantly less time for full training. $G_{SS}$ takes slightly longer iteration time (i.e, the runtime for each batch of training data) than $G$ due to the added ID passport layer, but its convergence time is significantly shorter than that of $G$. This is because $G_{SS}$ inherits the pretrained weights of $G$ before it is trained. Take BlendFace as an example, $G$ requires 300,000 iterations to converge, while $G_{SS}$ needs only 20,000 iterations to achieve convergence. Therefore, even though $G_{SS}$ has a slightly higher iteration time than $G$, its overall training time is around or more than an order of magnitude lower than that of $G$.

