# OpenReview forum: "Robust Secure Swap: Responsible Face Swap With Persons of Interest Redaction and Provenance Traceability"
_ICML.cc/2025/Conference — ICML 2025 poster_

### Official Review · Reviewer_tYBZ · 2025-03-17

**Overall Recommendation:** 3

**Summary:**

The work proposes a novel method to transfer a general face swap method to a secure face swap method, where POI is rejected and non-POI is passed to generate swapped face image with a tracable, unique, invisible watermark. Specifically, an ID Passport layer is proposed to recognize if the input face image is POI and a detachable watermark encoder and decoder  is trained to insert a tracable and invisible watermark into a swapped face of non-POI face image. Extensive experiments are conducted to demonstrate the effectiveness of the proposed method.

## update after rebuttal
After reading rebuttal, my final rating is weak accept.

**Claims And Evidence:**

N/A

**Essential References Not Discussed:**

N/A

**Experimental Designs Or Analyses:**

N/A

**Methods And Evaluation Criteria:**

N/A

**Other Comments Or Suggestions:**

N/A

**Other Strengths And Weaknesses:**

Strength
1. The task of secure swap is interesting and practically meaningful.
2. The paper is well-organized and -written, so that it is easy to follow.
3. The proposed method is technically reasonable.
4. The experiments are solid and present insightful analysis

Weakness
1. The ID passport layer is not clearly explained how to work.
2. How does IDConv perform? what is the structure of IDConv.
3. How to balance each item of loss functions to make sure the proposed network can work properly..

**Questions For Authors:**

N/A

**Relation To Broader Scientific Literature:**

N/A

**Theoretical Claims:**

N/A

---

> ### Author Rebuttal · Authors · 2025-03-31
>
> ### **(A) Other Weaknesses**
>
> - **A-Q1: ID passport layer.**
>
> **(I) Motivation**. Our design of the ID passport layer is driven by the goal of enabling ID-sensitive processing within the faceswap model. It is motivated by the need to differentiate outputs between POI and nonPOI based on ID information contained in the intermediate feature F. To this end, the feature F not only serves as input but also partially defines the convolutional parameters. This design enables the model to dynamically adapt the convolution operation according to ID.
> This design also encodes the watermark into the generated weights, enabling watermarking on non-POI without extra modules.
>
> **(II) Position**. The ID passport layer is placed at a later stage of the faceswap model, close to the output. In BlendFace, which contains 8 blocks, we inject it at the 7th block. This position has two advantages: (1) proximity to the output allows more direct manipulation in RGB space for watermark embedding; (2) features at this block have already captured rich ID-related information, which makes POI protection more effective.
> In contrast, shallow layers are less effective, as the distance from the output increases the difficulty of watermark embedding in RGB space, and the features are semantically weak in terms of identity, which compromises the ability for POI protection.
>
> **(III) Work pipeline**.
> The ID passport layer receives two inputs: feature F from the upstream network and the watermark, and outputs the final feature F'. As shown in **Fig. 3 in paper**, the layer is composed of multiple parallel ID-aware convolutions, denoted as IDConv1 to IDConvn. Details of IDConv are provided in A-Q2.
>
> - **A-Q2: The structure of each IDConv.**
>
> The structure of each IDConv is a layer of 3*3 convolution. The construction of the IDConv is presented in **Fig. 8 in the Appendix**.
>
> **(I)** Watermark encoder. The watermark encoder feeds input watermarks into a series of FC and ReLU layers to obtain $f$ watermark features $[wm_1,wm_2,…,wm_f]$.
>
> **(II)** Kernel generator. We feed the upstream feature $F\in \mathbb{R}^{c\times h\times w}$ into several convolution and ReLU layers to obtain a new feature $F1\in \mathbb{R}^{c’\times h’\times w’}$. Then $F1\in \mathbb{R}^{c’\times h’\times w’}$ goes through $n$ different convolution layers and generates $n$ different kernels of size $(f\times c’’\times h’’\times w’’)$. We utilize $[wm_1,wm_2,…,wm_f]$ to weight the $n$ kernerls for watermark embedding. Finally, we use $n$ different weighted kernels to initialize the $n$ IDConv.
>
> **(III)** Coefficient generator. It takes feature $F\in \mathbb{R}^{c\times h\times w}$ as input and passes it through the convolution and pooling layers. This is followed by the reshape operations to obtain n tensors of size $(c’\times h\times w)$, which are then transformed into n coefficients $[c_1, c_2,…,c_n]$ by the ReLU and fully connected layer. Finally, we use the $n$ coefficient $[c_1, c_2,…,c_n]$ to weight the n IDConv.
>
>
> - **A-Q3: Analysis of loss functions.**
>
> We balance multiple losses by assigning equal weights to each term. This is intentional and based on the following reasons:
> The watermark encoder, kernel generator, and coefficient generator have their learnable parameters, which enable our losses to achieve stable convergence without careful weighting. Besides, all components of the model are optimized jointly and simultaneously.
> The overall loss function consists of face-swapping losses (FS), a knowledge distillation loss (KD), a POI-specific loss, and a WM loss. FS and KD are applied to nonPOI samples, while the POI and WM losses act on ID protection and watermark embedding, respectively, which are non-conflicting. Equal weights are assigned to all components based on three observations:
>
> **(I)** Numerical scales remain comparable across losses;
>
> **(II)** FS and KD address similar objectives, and all losses operate on disjoint data partitions (nonPOI vs. POI);
>
> **(III)** The WM loss modifies only imperceptible features, preserving image quality and avoiding interference with other losses.
>
> We also acknowledge the importance of each loss component introduced to address a specific task:
>
> **(I)** FS loss and KD loss: enhances visual fidelity in face-swapped images.
>
> **(II)** POI loss: enforces protection of POI identities.
>
> **(III)** WM loss: embeds watermark into the output of nonPOI images.
>
> To evaluate their contributions, we took BlendFace as an example and conducted ablation studies by removing each loss independently in the table below. The results show significant performance degradation in the corresponding tasks on the removal of any loss term.
>
> |Loss|PSNR|SRmask|WM_Acc|
> |-|-|-|-|
> |w/o FS loss|32.3|0.93|0.98 |
> |w/o KD loss|27.7|1.00|0.99|
> |w/o POI loss|34.5|0.00|0.99|
> |w/o WM loss|35.2|1.00|0.51|

---

### Official Review · Reviewer_HFDi · 2025-03-24

**Overall Recommendation:** 3

**Summary:**

This paper introduces a method to prevent unauthorized face swaps involving persons of interest (POIs), while embedding an invisible watermark in non-POI results. Experiments demonstrate that the method maintains the performance of the original face swap model, effectively prevents unauthorized swapping, and ensures watermark-based provenance. The authors also conducted robustness tests under various attack scenarios.

**Claims And Evidence:**

Yes

**Essential References Not Discussed:**

No

**Experimental Designs Or Analyses:**

Yes. Tables 1 and 2 demonstrate that the model maintains the performance of the original face swap model. Tables 3 and 4 show the effectiveness of preventing unauthorized face swaps and successfully embedding watermarks. Figures 4, 5, and 6, along with Table 5, present the results under various attack scenarios.

**Methods And Evaluation Criteria:**

Yes

**Other Comments Or Suggestions:**

This is an interesting topic with potential applications across various domains—not only in face swapping, but also in talking head generation, digital humans, and related fields.

**Other Strengths And Weaknesses:**

The topic is both interesting and important in the field of security. This paper demonstrates that the proposed method can prevent unauthorized face swaps involving persons of interest (POIs) and embed an invisible watermark in non-POI results.

However, the paper could be improved in several areas. For instance, it does not include comparisons with other methods that address unauthorized face swaps involving POIs. Additionally, the performance improvement attributed to the watermarking technique appears marginal and may be influenced by randomness. Moreover, the flip accuracy for non-POI results under various image-level attacks is relatively low, which could potentially be improved through data augmentation strategies.

**Questions For Authors:**

Figure 5 shows the results of horizontal flipping on the augmented SecureSwap BlendFace model. However, it appears as a straight line, which differs significantly from the other curves—could the authors clarify the reason for this behavior?

**Relation To Broader Scientific Literature:**

This paper primarily focuses on the security aspect, while incorporating the face swap task, which is a form of image editing.

**Theoretical Claims:**

Yes. They are good.

---

> ### Author Rebuttal · Authors · 2025-03-31
>
> ### **(A) Other Weaknesses**
> - **A-Q1(I): Comparisons with other methods addressing unauthorized face swaps and watermark.**
>
> **Compare with anti-faceswap methods.** Existing methods to address unauthorized face swaps fall into two categories: proactive protection and post-hoc detection. Post-hoc detection suffers from inherent latency and cannot effectively prevent image misuse. Current proactive methods primarily rely on adversarial attacks by adding perturbations to protected images, such as POI photos, to prevent forgery [1][2][3]. These approaches offer image-level protection, require costly preprocessing, and cannot scale to large volumes. In contrast, our work targets identity-level protection. Given the one-to-many nature between identity and images, our method supports scalable protection without preprocessing and achieves higher robustness. As suggested, the following table compares our defense success rates (SRmask) against existing methods when protecting 128 images with different identities.
>
> | Model| Disrupting Deepfakes[1]| Initiative Defense[2] | CMUA[3] | Ours |
> |:-:|:-:|:-:|:-:|:-:|
> | SimSwap | 0.93 | 0.94  | 0.99 | 1.00 |
> | FaceShifter | 0.91 | 0.90  | 0.94 | 1.00 |
> | BlendFace  | 0.85 | 0.84 | 0.89 | 1.00 |
> | MobileFSGAN   | 0.90  | 0.92 | 0.95 | 1.00 |
>
> **Compare with watermark methods.** Unlike existing watermarking methods (training data based embedding [4], watermark decoder based supervised embedding [5,6]), our method directly encodes the watermark into model parameters. Although the watermark is also extracted from images, it becomes part of the model parameters. Once initialized by a watermark, the model no longer requires external watermark input during inference. This supports efficient, scalable creation of uniquely watermarked model instances. Our design allows rapid deployment of distinct watermarked model instances at scale.
>
> [1] Disrupting Deepfakes: Adversarial Attacks Against Conditional Image Translation Networks and Facial Manipulation Systems
> [2] Initiative Defense against Facial Manipulation
> [3] Cmua-watermark: A cross-model universal adversarial watermark for combating deepfakes
> [4] Artificial fingerprinting for generative models: Rooting deepfake attribution in training date
> [5] Wide flat minimum watermarking for robust ownership verification of gans
> [6] The stable signature: Rooting watermarks in latent diffusion models
>
> - **A-Q1(II): Watermark performance improvement and stability.**
>
> Our watermarking method demonstrates promising performance in both effectiveness and usability, as shown in Figure 4 in our main paper and Figure 9 in the Appendix.
> In terms of efficiency, by binding the watermark to model weights, our method enables scalable watermarking for efficient deployment in distribution scenarios.
> To address the potential influence of randomness, we report the average accuracy and standard deviation over 1,000 different embedded watermarks, as shown in the table below. The low standard deviations indicate consistent performance across watermark instances. These results suggest that our method remains robust and is not affected by randomness.
>
> |Model|BlendFace|SimSwap|FaceShifter|MobileFSGAN|
> |-|-|-|-|-|
> |Acc|99.48|99.93|100.0|98.24|
> |std|0.33|0.29|0.00|0.30|
>
> - **A-Q1(III): Flip accuracy improvement by suggested data augmentation strategies.**
>
> According to your suggestion, we used a stronger data augmentation to improve the watermark robustness against horizontal flip attack.
> Specifically, we increased the probability of applying horizontal flipping to 30% in each training step, and evaluated the average watermark accuracy on 1k CelebA images, as shown in the table below. It demonstrates that flip accuracy has improved after applying data augmentation. Additionally, to explore the impact of image quality, we calculated the PSNR between the watermarked and non-watermarked images. Results in the table below show that augmentation did not introduce any noticeable degradation in image quality. Overall, thanks to the reviewers' suggestion, we adopt data augmentation. This strategy improves flip accuracy.
>
> | Method| SimSwap | FaceShifter | BlendFace |MobileFSGAN |
> |-|-|-|-|-|
> | Horizontal Flip|93.7|93.9|94.1|93.7
> | PSNR| 33.7| 34.7 | 34.3 | 34.1
>
> ***
>
> ### **(B) Questions For Authors**
>
> - **B-Q1: Clarification of horizontal flipping appears as a straight line.**
>
> In our experiments, unlike Gaussian noise or adversarial perturbations, which can be applied at varying levels, flipping is a binary transformation. Thus each image has only two possible states: flipped or not flipped, without any gradual increasing attack intensity.  Consequently, the result is shown by a straight line in our evaluation, rather than a curve that reflects increasing attack strength. We will clarify this in the paper to avoid potential misunderstandings.

---

> > ### Comment · Reviewer_HFDi · 2025-04-03
> >
> > This is an interesting topic with potential applications across various domains—not only in face swapping, but also in talking head generation, digital humans, and related fields.

---

> > > ### Author Response · Authors · 2025-04-05
> > >
> > > Dear Reviewer HFDi,
> > >
> > > First, we would like to sincerely thank you for your valuable review.
> > >
> > > We are grateful for your questions about our watermarking method and presentation, which helped us improve the clarity of our design. In addition, your suggestion to explore data augmentation was helpful. We followed it in our experiments. The results confirmed its effectiveness and further strengthened the persuasiveness of our work.
> > >
> > > We particularly appreciate your recognition of the topic of our work. Your comment — *'This is an interesting topic with potential applications across various domains—not only in face swapping, but also in talking head generation, digital humans, and related fields.'* — is very encouraging and reinforces our confidence in our work.
> > >
> > > We are grateful for your recognition of our rebuttal. We would sincerely appreciate it if this could be reflected in a more clear positive rating. We fully respect your judgment and thank you again for your valuable feedback.

---

### Official Review · Reviewer_JXWv · 2025-03-24

**Overall Recommendation:** 3

**Summary:**

The paper presents a method that incorporates a trainable adapter into an existing GAN-based face-swapping pipeline to safeguard the privacy of Persons of Interest (POIs) by redacting their appearance in the output. Additionally, it embeds a watermark for traceability while preserving identity transferability for non-POI swaps. Experimental results demonstrate the effectiveness of the method in achieving POI redaction, watermark embedding, and robustness against potential attacks.

**Claims And Evidence:**

The two major claims—POI redaction and watermark embedding without compromising output quality—are well-supported, except for some ambiguity in POI redaction when different POI images are used as model inputs during inference.

**Essential References Not Discussed:**

The essential references are appropriately discussed to provide a clear understanding of the contributions. However, further discussion is needed on the difference between previous generative model watermarking techniques and the approach presented in this paper. (Lines 73-76 [Right] seem to refer to POI redaction, not watermarking, I believe.)

**Experimental Designs Or Analyses:**

1.	Did you test POI redaction with diverse images, such as a celebrity’s photo under dark lighting, with a different hairstyle, or at a slightly different age? Can the model still effectively redact these variations?
2.	A common face-swapping evaluation metric is FID (Fréchet Inception Distance), which measures output image fidelity/quality. How does adding the watermark impact FID?
3.	Did you evaluate the False Positive Rate (FPR) of the redaction? Specifically, what percentage of redaction occurs when a non-POI is used as the source image?

**Methods And Evaluation Criteria:**

1.	The FFHQ dataset includes face images with extreme poses and varying ages. It would be beneficial to show results on this dataset as well to evaluate whether the method performs consistently (Both with training and without training).
2.	How long does it take to train the model to redact a new POI? Additionally, why not simply compare the incoming source image with stored POI images using ID similarity (e.g., ArcFace) and remove matches based on a threshold? While this approach requires storing images, wouldn’t it be more efficient than lengthy model training?
3.	The method incorporates numerous loss components, but their individual significance is not thoroughly analyzed.
4.	The motivation behind the design choices and the positioning of the ID passport layer within the face-swapping model remain unclear.

**Other Comments Or Suggestions:**

1.	Correct the spellings (E.g Tab 4, Ln262 [right]  Arrtributing-> Attributing)
2.	Table 4 can be summarized in one sentence as everything are 1.
3.	y_{s,t} -> x_{s,t} in eq.3,4.

**Other Strengths And Weaknesses:**

Strengths:
        1.	The task is valuable to the research community.
	2.	The experiments are extensive.

Other Weaknesses:
	1.	Figure 1 is not referenced in the text.
	2.	The works in the tables should be properly cited.
	3.	In Table 3, are the POI IDs 128, 512, and 1024 for SimSwap?

**Questions For Authors:**

1.	Why isn’t this method suited for diffusion-based face swapping? Diffusion models are a powerful alternative to GAN-based methods, offering excellent image quality and are currently a leading generative tool. Could this method be integrated into diffusion U-Nets?
2.	Did you test POI redaction with various images during inference, such as a celebrity’s photo under dark lighting, with a different age, or hairstyle? Can the model still redact these variations effectively? (This was previously asked)
3.	How long does it take to train the model to add a few new POIs? What impact does this have on the redaction of existing POIs? For instance, if A, B, and C are the previous POIs and now you need to add X, Y, and Z, will redaction for A, B, and C still be valid? A simpler experiment could involve adding PubFig83 followed by VGGFace2 as POIs and validating on both datasets.
4.	What are the results on an untrained dataset, such as directly inferring on FFHQ? I want to understand the impact of the Robust Secure Swap method on the generalizability of the original face-swapping model.

**Relation To Broader Scientific Literature:**

The idea is valuable for addressing privacy concerns (POI redaction) directly during face-swapping generation, unlike most existing methods that apply redaction after the forged image has already been created.

**Theoretical Claims:**

There are no theoretical claims.

---

> ### Author Rebuttal · Authors · 2025-03-31
>
> ### **(A) Method & Evaluation Criteria**
> - **A-Q1: Performance on FFHQ.**
> We did not consider FFHQ as it lacks ID labels to support direct POI reduction evaluation. Still, we performed evaluation via data augmentation. To evaluate FFHQ as nonPOI, we trained models with FFHQ and calculated the quality difference between Gss and G outputs, as shown in columns 2 to 5 of the table below. We also present the results of directly inferring on FFHQ without training, as shown in columns 6 to 9. To evaluate FFHQ as POI, we randomly selected 128/512/1024 images from FFHQ, treating each image as a POI. For each POI, only one image was used for training, and 100 augmented samples were generated for test. Results (columns 10 to 12) show that even with one image per POI for training, we achieve over 85% protection success.
>
> |Model |PSNR |SSIM|LPIPS |FID|PSNR'|SSIM'| LPIPS'|FID'|POI=128|POI=512|POI=1024|
> |:-:|:-:|:-:|:-:|:-:|:-:|:-:|:-:|:-:|:-:|:-:|:-:|
> |SimSwap|34.60|0.97|0.01|15.67|34.11|0.97|0.01|15.21|0.98|0.92|0.87 |
> |FaceShifter|34.23|0.97|0.03|17.42|33.98|0.97|0.03|17.05|0.98|0.90|0.89|
> |BlendFace|34.24|0.96|0.02|12.81|34.08|0.96|0.03|12.59|0.97|0.91|0.88|
> | MobileFSGAN|34.27|0.95|0.01|16.32|33.89|0.96|0.02|15.80|0.94|0.89|0.86|
> - **A-Q2: ArcFace with threshold.**
> For time cost of new POI, see D-Q3.
> ArcFace-based defense is vulnerable to model-level attacks, as it didn't modify the model.
> Once the matching is bypassed, protection fails.
> In contrast, our method is robust and any model-level attacks would destroy generation capability (Fig. 9 in Appendix).
> Moreover, threshold-based methods suffer from a fixed threshold, which may result in high FPR or FNR if chosen improperly.
> - **A-Q3: Loss analysis.** See Review tYBZ A-Q3
> - **A-Q4: ID passport layer.** See Review tYBZ A-Q1&2
> ***
> ### **(B) Experimental Design and Analyses**
> - **B-Q1: Diverse POI images.**
> VGGFace2 and PubFig83 (evaluated in our paper) vary in illumination, hairstyles, poses, age, etc. Based on the results, we consistently achieve protection effectiveness across these diverse conditions.
> - **B-Q2: Watermark performance.**
> We reported 3 FID scores on CelebA in the table below: FID from baseline model G, FID from our method, and FID from our method with watermark only (no POI defense). Our method caused a small FID increase of around 1, which remains visually acceptable. When POI defense is removed and only watermark is applied, the FID increase is marginal and stays within a 1-point range compared to baseline, indicating that our method has a small impact on the image in both POI reduction and watermark.
>
> |Model|G (no WM)|Gss (WM + POI)|Gss (only WM)|
> |:-:|:-:|:-:|:-:|
> |SimSwap|12.95|14.52|13.12|
> |FaceShifter|14.67|15.98|15.34|
> |BlendFace|10.58|11.20|11.07|
> |MobileFSGAN|13.96|14.65|14.35|
>
> - **B-Q3: FPR.**
> We evaluated the FPR of POI redaction. Our evaluation consistently yields a FPR of 0: throughout our experiments, we did not observe any failure cases in which a nonPOI sample was mistakenly treated as a POI and subjected to redaction. We determine success using SRmask (by Eq. 15 in Appendix, with the threshold=0.05).
>
> ***
>
> ### **(C) Essential References**
> See Review HFDi A-Q1
> ### **(D) Questions for Authors**
> - **D-Q1: Adaptation to DM.**
> Our method can be applied to diffusion based faceswap models. We can insert the ID passport layer into the decoder (rather than the UNet) and finetune the decoder to prevent POI generation. We evaluated this design on DiffSwap [1]. Table below reports the protection performance and fidelity for protecting 128 POI with 16/32 training images, which shows unaffected image quality and near-perfect protection with only 32 training images per POI.
>
> |Metric|128(16)|128(32)|
> |-|-|-|
> |SRmask|0.975|0.996|
> |LPIPS|0.019|0.019|
>
> [1] DiffSwap: High-Fidelity and Controllable Face Swapping via 3D-Aware Masked Diffusion
> - **D-Q2: Diverse POI images.**
> See (B) B-Q1
> - **D-Q3: New POI.**
> Since the model has already converged on old POI(1024) and watermark, only a little finetuning is needed. Table below shows the computation time for 10 or 100 new POI, when old POI is or not involved.
>
> |Model|1024+10 (old POI involved)|1024+100 (old POI involved)|1024+10 (old POI not involved)|1024+100 (old POI not involved)|
> |-|-|-|-|-|
> |SimSwap|~1h 20m|~2h40m|~40m|~1h10m|
> |FaceShifter|~1h10m|~2h20m|~1h|~1h15m|
> |BlendFace|~1h10m|~2h20m|~50m|~1h5m|
> |MobileFSGAN|~1h10m|~2h|~35m|~1h10m
>
> The table below shows SRmask score (old/new) on old and new POI, with or without old POI included during finetuning. Results show that once old POI are involved in finetuning, their protection remains effective.
> |Model|1024+10 (old POI involved)|1024+100 (old POI involved)|1024+10 (old POI not involved) | 1024+100 (old POI not involved)|
> |-|-|-|-|-|
> |SimSwap|1.0/1.0|1.0/1.0|0.76/1.0|0.62/1.0|
> |FaceShifter|1.0/1.0|1.0/1.0|0.68/1.0|0.60/1.0|
> |BlendFace|1.0/1.0|1.0/1.0|0.75/1.0|0.69/1.0|
> |MobileFSGAN|1.0/1.0|1.0/1.0|0.72/1.0|0.65/1.0
> - **D-Q4: FFHQ.**
> See (A) A-Q1

---

> > ### Comment · Reviewer_JXWv · 2025-04-03
> >
> > Rebuttal answers most of my questions. Upon acceptance, please release the code publicly.

---

> > > ### Author Response · Authors · 2025-04-05
> > >
> > > Dear Reviewer JXWv,
> > >
> > > First, we deeply appreciate your time and effort on reviewing our paper.
> > >
> > > We would like to express our sincere gratitude for your positive evaluation and the initial weak accept recommendation. Beyond raising insightful concerns, you kindly guided us on how to design further validation experiments, which we found extremely helpful.
> > >
> > > Following your suggestions, we extended our experiments and obtained new results that, we believe, significantly enhance the clarity and technical soundness of our paper. We are also glad that our rebuttal was able to address your concerns. These new experiments will be incorporated into both the main paper and the appendix in the revised version.
> > >
> > > Moreover, we fully agree with your suggestion to release the code upon acceptance. We take this as an encouraging sign of your support for our work.
> > >
> > > In light of this, we would deeply appreciate it if you would consider updating to a more positive score. Nevertheless, we fully respect your judgment regardless of the final score, and again, we thank you for your constructive review.

---

### Decision · Program_Chairs · 2025-05-01

**Decision:**

Accept (poster)

**Comment:**

The paper received all accept recommendations. AC agrees with this recommendation and therefore is happy to accept the paper. Please address reviewer comments to the camera-ready version of the paper.